# Three Soil Bacterial Communities from an Archaeological Excavation Site of an Ancient Coal Mine near Bennstedt (Germany) Characterized by 16S r-RNA Sequencing

Linda Ehrhardt [1,2], P. Mike Günther [1,2], Manfred Böhme [3], J. Michael Köhler [1,2,*] and Jialan Cao [1,2]

1 Department of Physical Chemistry and Microreaction Technologies, Institute of Chemistry and Biotechnology, Technical University Ilmenau, Prof.-Schmidt-Str. 26, D-98693 Ilmenau, Germany

2 Institute of Chemistry and Biotechnology, Technical University Ilmenau, D-98684 Ilmenau, Germany

3 Landesamt für Denkmalpflege und Archäologie Sachsen-Anhalt, D-06020 Halle/Saale, Germany

* Correspondence: michael.koehler@tu-ilmenau.de

**Abstract:** This metagenomics investigation of three closely adjacent sampling sites from an archaeological excavation of a pre-industrial coal mining exploration shaft provides detailed information on the composition of the local soil bacterial communities. The observed significant differences between the samples, reflected in the 16S r-RNA analyses, were consistent with the archaeologically observed situation distinguishing the coal seam, the rapidly deposited bright sediment inside an exploration shaft, and the topsoil sediment. In general, the soils were characterized by a dominance of *Proteobacteria*, *Actinobacteria*, *Acidobacteria*, and *Archaea*, whereas the coal seam was characterized by the highest proportion of *Proteobacteria*; the topsoil was characterized by very high proportions of *Archaea*—in particular, *Nitrosotaleaceae*—and *Acidobacteria*, mainly of Subgroup 2. Interestingly, the samples of the fast-deposited bright sediment showed a rank function of OTU abundances with disproportional values in the lower abundance range. This could be interpreted as a reflection of the rapid redeposition of soil material during the refilling of the exploration shaft in the composition of the soil bacterial community. This interpretation is supported by the observation of a comparatively high proportion of reads relating to bacteria known to be alkaliphilic in this soil material. In summary, these investigations confirm that metagenomic analyses of soil material from archaeological excavations can provide valuable information about the local soil bacterial communities and the historical human impacts on them.

**Keywords:** soil ecology; bacterial communities; coal mining; archaeology; metagenomics; ecological memory; human impact; extremophiles





## 1. Introduction

Natural soils contain a huge number of different microorganisms. Many of them are known from genomic studies, but most soil microorganisms have not yet been cultivated [1]. The composition of soil bacterial communities is strongly dependent on chemical factors, such as pH, salt content, usable substrates, and humidity [2–4]. In addition, noxious substances can suppress the growth of a large number of soil microbes; these substances include pesticides, plant fertilizers and other agrochemicals, and heavy metal ions [5–7].

It is well known that human activities have a strong effect on soil fertility and microbial composition. Industry and mining are especially strong factors, as they are often associated with the release of soil-altering wastes. In particular, the acidic effluents from mines, smelters, and slag deposits cause serious environmental damage [8–10]. Not only does the damage occur directly from lowered pH, but acidification is also frequently related to increased erosion, chemical decomposition of minerals, and mobilization of toxic metals. Potash mines release large amounts of salt, causing high levels of salinity in the soil surrounding the mines—especially in valley areas—as well as in the rivers of the concerned

regions. However, despite the negative impacts of human engineering activities on microorganisms, such as through mining [11–14], soil fertility, and ecological robustness [15–17], technical environments can also contribute to the microbial diversity of microbiological populations. There are many cases of researchers reporting the discovery of new bacteria, for example, in special technical environments. In addition to bioreactors, other specialized equipment settings have also been sources of newly described bacterial species and interesting microbial metabolic features [18,19].

Mining activities can have a strong impact on the development of soils and soil bacterial communities. This is not only relevant for active mining areas but also the post-mining period [20]. Of particular interest is the fact that human activities not only damage soils but can also contribute to local soil bacterial diversity. Indeed, the beta diversity can be increased by local changes in the growth and conservation conditions of soil bacteria [21]. Archaeology is increasingly becoming a focus of genetic investigations [22–24]. Thus, special soil bacterial communities have been reported in archaeological excavation sites—prehistoric buildings, settlement places, and tombs [25–30]—and medieval or early modern pre-industrial mining and smelting sites [31–34]. Despite the investigations of soil bacterial communities on modern and historical metal mines and smelting sites, less attention has been paid to historic coal mines. A study of US coal mines showed that the impact of metal release from coal mines is underestimated [35].

In the frame of an archaeological program studying ancient mining places, a historical coal mine near Bennstedt (Saalekreis, Sachsen-Anhalt, Germany) was investigated. Samples were taken from the topsoil, a sediment layer, and the rim of the coal seam. Herein, the results of a comparative 16S r-RNA study of the soil bacterial communities of these three different sampling sites are reported.

## 2. Experimental

### 2.1. Excavation Site and Soil Samples

Soil samples were taken as part of an archaeological investigation of an ancient lignite mining area near Bennstedt. The old exploration mining site is marked by comparatively small shafts perpendicular to the coal seam. After completion, the shaft hole was backfilled comparatively quickly with bright sediment. Later, the residual depression of the shaft hole backfill was gradually filled with topsoil (brown sediment). An overview of the construction area and the archaeological excavation site is shown in the Supplementary Materials (Figure S1).

The sampling sites were chosen close to each other in an archaeological section of an ancient exploration shaft (Figure 1), which was probably constructed in the second half of the 18th century or around 1800. The sampling site is located about one kilometer northeast of Bennstedt (GK coordinates: E 4488789/N 5706398). Two samples for independent DNA extraction (HB57-1 and HB57-2) originated directly from the coal seam. Two other samples were from the primary deposited bright sediment (HB58-1, HB58-2), and two further samples (HB59-1 and HB59-2) were from the topsoil filling material of the upper part of the residual hole trough. All samples were characterized by a rather low pH and moderate to low electrical conductivity (Table 1). The measurements of electrical conductivity indicated a slightly increased salt content of the coal layer in comparison with the sediments.

**Table 1.** pH and electrical conductivity of the sampling sites.

|  |  | Depth | pH | Conductivity |
|---|---|---|---|---|
| HB57-1, HB57-2 | coal seam | 1.6 m | 4.22 | 114.8 (µS/cm) |
| HB58-1, HB58-2 | bright sediment | 1.6 m | 4.09 | 43.3 (µS/cm) |
| HB59-1, HB59-2 | topsoil sediment | 1.0 m | 4.05 | 45.2 (µS/cm) |

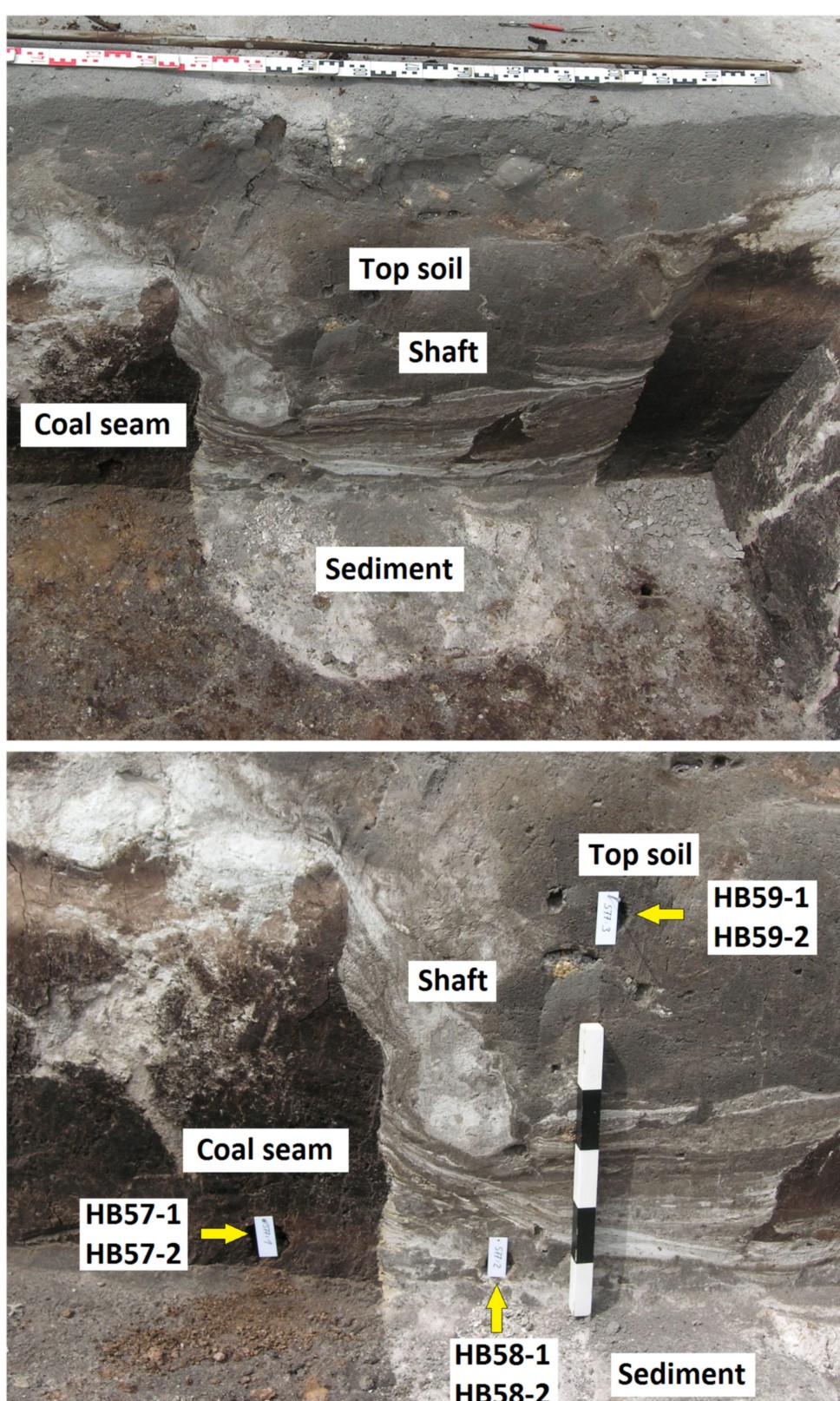

**Figure 1.** Sampling sites. The picture shows the three different structures of an exploratory shaft of an ancient lignite mining site. Two samples were taken from each layer (topsoil, coal seam, and sediment). The depth and width can be estimated using the reference bars. The black and white bar is 50 cm long in total (picture below), and the other has 5 cm per section (picture above).

## 2.2. DNA Extraction, Amplification, and Labelling

DNA was isolated from the soil samples using DNeasy® PowerSoil® Pro kits (Qiagen, Hilden, Germany). The kits were used according to the manufacturer's instructions. The laboratory thermocycler Edvocycler (Edvotek, Washington, DC, USA) was used for polymerase chain reactions (PCR). The quality of PCR products was checked by gel electrophoresis with 1% agarose gels. This test was performed after each PCR. The primary PCR products as well as the final pooled libraries were purified using the ProNex® Size-Selective Purification System (Promega, Madison, WI, USA) according to the supplier's protocol.

Adaptor primers supplied by Eurofins Genomics (Ebersberg, Germany), Amplicon PCR A519F-Ad (5′ TCGTCGG-CAGCGTCAGATGTGTATAAGAGACAGCAGCMGCCGC GGTAA 3′) and Bact_805R-Ad (5′-GTCTCGTGGGCTCGGAGATGTGTATAAGAGACAGG ACTACHVGGGTATCTAATC 3′), were used at a concentration of 100 pmol/μL. The PCR mixtures (25 μL in total per reaction) were composed as follows: 0.5 μL of DNA isolation eluate, 2 mM $MgCl_2$, 200 μM dNTP mix, 0.65 Units GoTaq® G2 Hot Start DNA Polymerase, nuclease-free water (all reagents from Promega, Madison, WI, USA), and 1 μM of each primer. For PCR amplification, the following steps were performed: initial denaturation for 5 min at 94 °C, followed by 30 amplification cycles involving 30 s denaturation at 94 °C, 30 s primer annealing at 50 °C, and 30 s extension at 72 °C. The temperature cycles were finished with a final extension reaction at 72 °C for 5 min.

The forward and reverse indexing primers used for index PCR were also supplied by Eurofins Genomics (Ebersberg, Germany). These primers were used at a concentration of 1.25 pmol/μL. For index PCR with a total volume of 25 μL per reaction, the following composition was applied: 2.5 μL amplicon PCR product, 2.5 mM $MgCl_2$, 300 μM dNTP mix, 0.5 units GoTaq® Mdx Hot Start DNA Polymerase, nuclease-free water (all reagents from Promega, Madison, WI, USA), and 125 nM of each of the two primers of the respective indexing primer pair.

For index primer PCR, the following program steps were applied: initial denaturation for 3 min at 95 °C, followed by 30 amplification cycles involving 30 s denaturation at 95 °C, 30 s primer annealing at 55 °C, and 30 s extension at 72 °C. The temperature cycles were finished with a final extension at 72 °C for 5 min.

## 2.3. Processing of NGS Data

The obtained fastq files of forward- and reverse-aligned 16S rRNA data were first converted to fasta format contig files and quality files (mothur (version 1.39.5)) using the open-source Galaxy platform (https://usegalaxy.org; accessed on 19 January 2022). All investigated datasets were characterized by high median quality scores.

The contig files were aligned to rRNA databases based on the NCBI cloud using the SILVAngs data analysis service (https://ngs.arb-silva.de/silvangs; accessed on 21 January 2022). This allowed a detailed community analysis of previously obtained sequencing data [36–38]. For all analyses, the preset parameter configurations of SILVAngs database version 138.1 were applied [38].

In most cases, the sequencing data allowed assignment to taxonomic groups down to the genus level. In some other cases, it was only possible to identify higher taxonomical levels, such as families, orders, classes, or phyla. The lowest identified level for each distinguished bacterial type is referred to as the Operational Taxonomic Unit (OTU). The compositions of soil bacterial communities were based on OTUs, which means they were based on the genera as much as possible.

## 2.4. Data Analyses

Different methods were used to evaluate the taxonomic data and sample-specific abundances of OTUs.

First, the percentages of the most abundant phyla were simply compared with a bar chart. The associated plot included an automatic normalization of the read counts. Bar

charts were also used to plot the absolute read counts reflecting the different abundances of specific OTUs in various samples.

On the one hand, the correlation of abundances when considering the complete data sets was represented by binary logarithmic correlation plots. These allowed the comparison of sample pairs on the basis of the individual abundances of all detected OTUs by plotting these abundances with one sample as the x-axis and the other as the y-axis. On the other hand, correlation coefficients were calculated for all sample pairs. The complete table can be found in Table S1.

Since double logarithmic correlation plots give a better impression of the relationship between pairs of samples than linear plots, normalized logarithmic abundance values were included in some analyses. These normalized abundance r-values are expressed as the ratio of the individual read counts N to the total read count of a sample $N_{sum}$:

$$r = \log10 \, (1 + 10^6 \times N/N_{sum}) \tag{1}$$

In these graphs, the complete populations in the bacterial communities are compared according to the magnitude of abundances. In this way, the correlations between more and less abundant OTUs are presented in a common picture.

The distribution of OTUs in the different samples was also investigated by principle component analyses (PCAs). The corresponding two-dimensional plots illustrate the similarities and dissimilarities of samples very clearly.

Finally, rank diagrams were also used for a general comparison of the distribution of abundances in each sample. In this approach, all OTUs in a sample were ranked according to their number of reads. The differences in the distribution of OTU abundances could be readily illustrated by these plots, which represent the data sets as normalized rank plots.

The difference between the bright sediment communities and the other samples was illustrated by rank-difference plots between the real and theoretical ideal distribution of abundances. Therefore, the logarithmic rank plots were approximated by a linear function reflecting the overall exponential character of the abundance distribution. This approximation is given by the following equation:

$$N_0 = e^{\,a \times k} \qquad or \qquad \ln \, (N_0) = a \times k \tag{2}$$

where $N_0$ describes the (theoretical) number of reads for each OTU corresponding to its position k in the rank order. The coefficient a depends on the ratio of the total number of reads to the OTUs.

## 3. Results and Discussion

### 3.1. Comparison of Sampling Sites According to the Bacterial Soil Community at Phylum Level

All samples contained a significant amount of *Archaea* and were characterized by comparatively high levels of *Proteobacteria* and *Acidobacteria*. The topsoil showed a particularly high content of *Archaea* and *Acidobacteria*. The highest proportion of *Proteobacteria* was found in the samples from the coal seam (HB57-1 and HB57-2).

Even stronger differences were observed in some other phyla (Figure 2). Thus, the coal seam samples were characterized by a significant proportion of *Myxococcota*. By contrast, the bright sediment and topsoil samples contained less *Myxococcota* but a considerable proportion of *Firmicutes*. Both topsoil samples examined had significantly lower proportions of *Verrucomicrobia* than the coal seam and bright sediment samples. All the phyla mentioned were typical of soil samples. However, their different ratios in the sample content reflected characteristic differences in the respective bacterial communities in the soil.

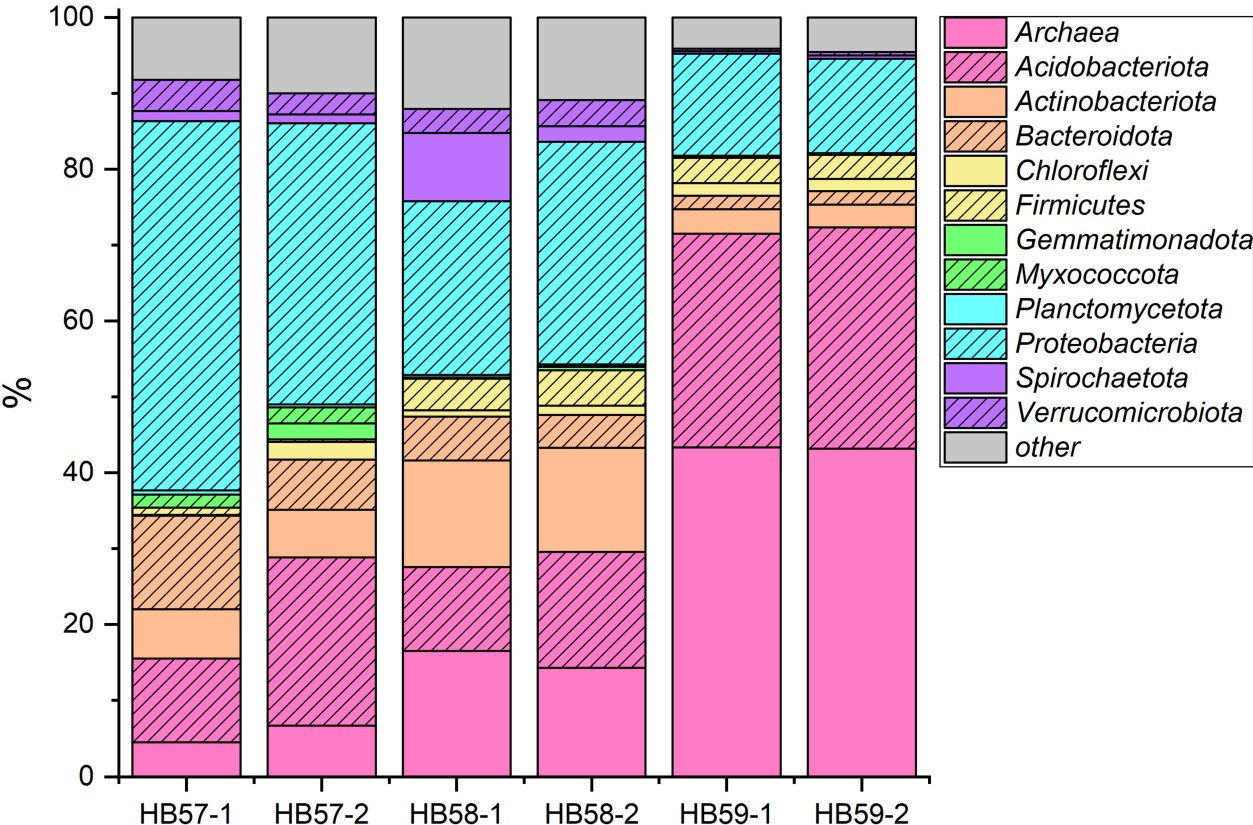

**Figure 2.** Stacked column plot of percentage distribution of the soil microbiome at phylum level. Each sample, including duplicate sample locations, is shown in a separate column. The 12 most dominant phyla are named. Low-abundance representatives are summarized in others.

### 3.2. Rank Profiles of OTU Abundances

The analysis of soil bacterial community composition at the OTU level provided much more information about the characteristics of the three closely adjacent but distinct microbial habitats. The sequencing data supplied between 252 and 281 OTUs with at least one read in each of the six samples; between 119 and 161 OTUs had at least 10 reads. Greater differences in the number of reads per sample were observed in OTUs with a large number of reads. By contrast, only 8 and 10 OTUs with more than 1000 reads were found in the topsoil (HB59-1 and HB59-2); 21 OTUs with more than 1000 reads were found in the coal seam samples (HB57-1 and HB57-2). The two samples of bright sediment yielded 19 and 31 OTUs with more than 1000 reads.

The distribution of OTU abundances was compared using normalized rank plots (Figure 3). They illustrated that, on the one hand, the topsoil was characterized by a few OTUs with a very high number of reads (inset in Figure 3). On the other hand, the samples of bright sediment filling the shaft (HB58-1 and HB58-2) were characterized by comparatively high abundances of less frequent OTUs. The frequency distribution of OTUs from the coal seam samples was between that of the other two pairs of samples. Plots of the empirically identified rank function and the linearly approximated plots of the logarithmic rank diagrams are shown in the Supplementary Materials (Figure S2). The most abundant OTUs were over-represented in all samples compared with this linear function. On the one hand, these most abundant OTUs represented the most active bacteria in the soil. On the other hand, the decreasing number of reads in the rank order referred to less active or dormant bacteria, which may have been more active in the past. It is assumed that the continuous decrease in the number of reads, closely following the exponential ranking function for the less abundant OTUs, is the standard case.

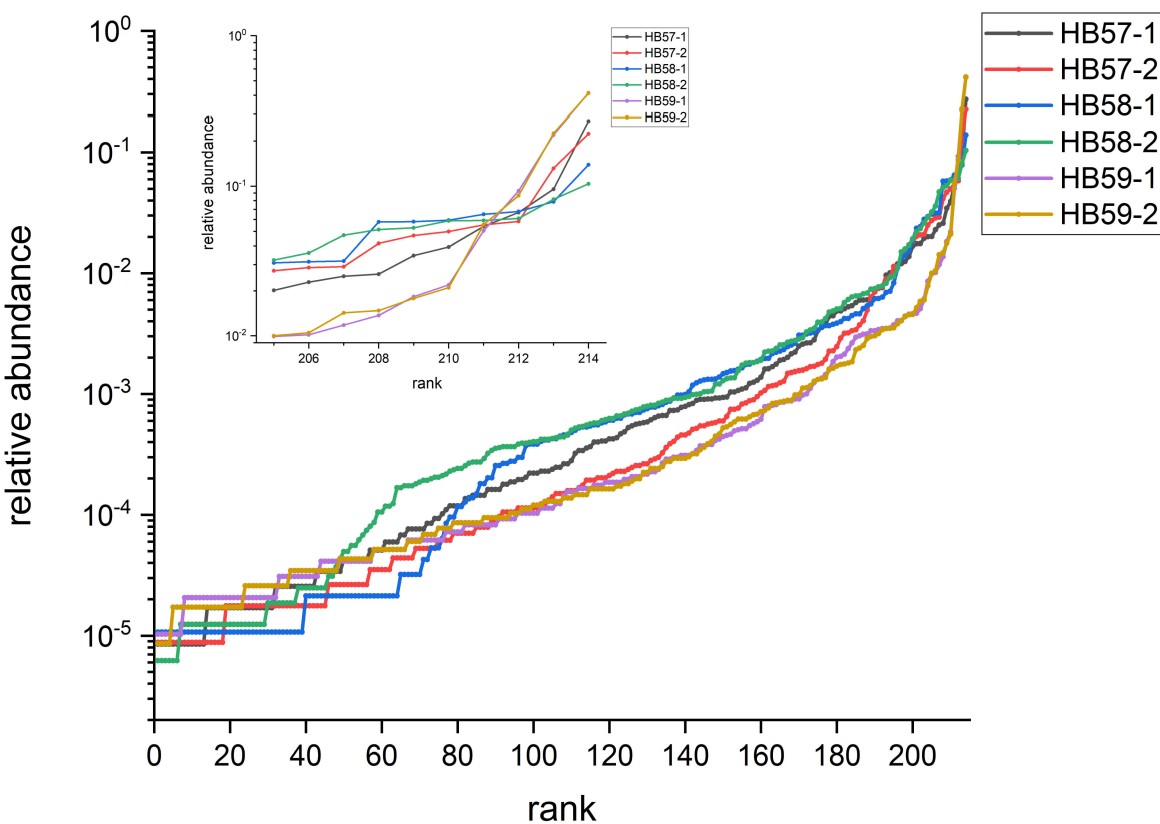

**Figure 3.** Normalized rank plots (inset: normalized frequencies of the top ten OTUs for each sample). The y-axis is a logarithmic scale. It shows the relative frequency (y-axis) of the ranked OTUs from low read counts (lower rank, x-axis) to high read counts (upper rank, x-axis). Therefore, the rank around 230 represents the most frequent OTU from the NGS analysis. However, for each sample, this may mean a different species.

The differences between the ranking functions became clearer by plotting the ratio of empirical and theoretical numbers in the ranking (Figure 4). The plots show the very high importance of the most abundant OTUs of the topsoil samples (Figure 4e,f). In addition, they very clearly display an overrepresentation in the group of less abundant OTUs in the samples from the bright sediment (Figure 4c,d), which was not found in the other soils. It is remarkable that this overrepresentation of less abundant species in samples HB58-1 and HB58-2 was combined with a lower overrepresentation of the most abundant OTUs. It is discussed here that the specific shape of this function should not be considered as an occasional abundance distribution; rather, it is the consequence of a specific history of the soil bacterial community. The positive deviation in the frequency distribution of the less abundant OTUs found in the bright sediment could have been caused by a massive change in the environmental conditions in the past. One possible reason could have been the quick relocation of soil material with highly active OTUs to a less suitable new environment. An obvious reason for this change could be the rapid filling of the shaft hole after its delivery. This interpretation would mean that the characteristic maxima in Figure 4c,d indicate that the soil material was relocated in an event that took place more than 200 years ago.

### 3.3. Comparison of Sampling Sites by Special OTUs

In this section, the abundances of the individual OTUs are regarded. They were compared either by the absolute number of reads N or by the normalized logarithmic value r (Equation (1), Section 2.4).

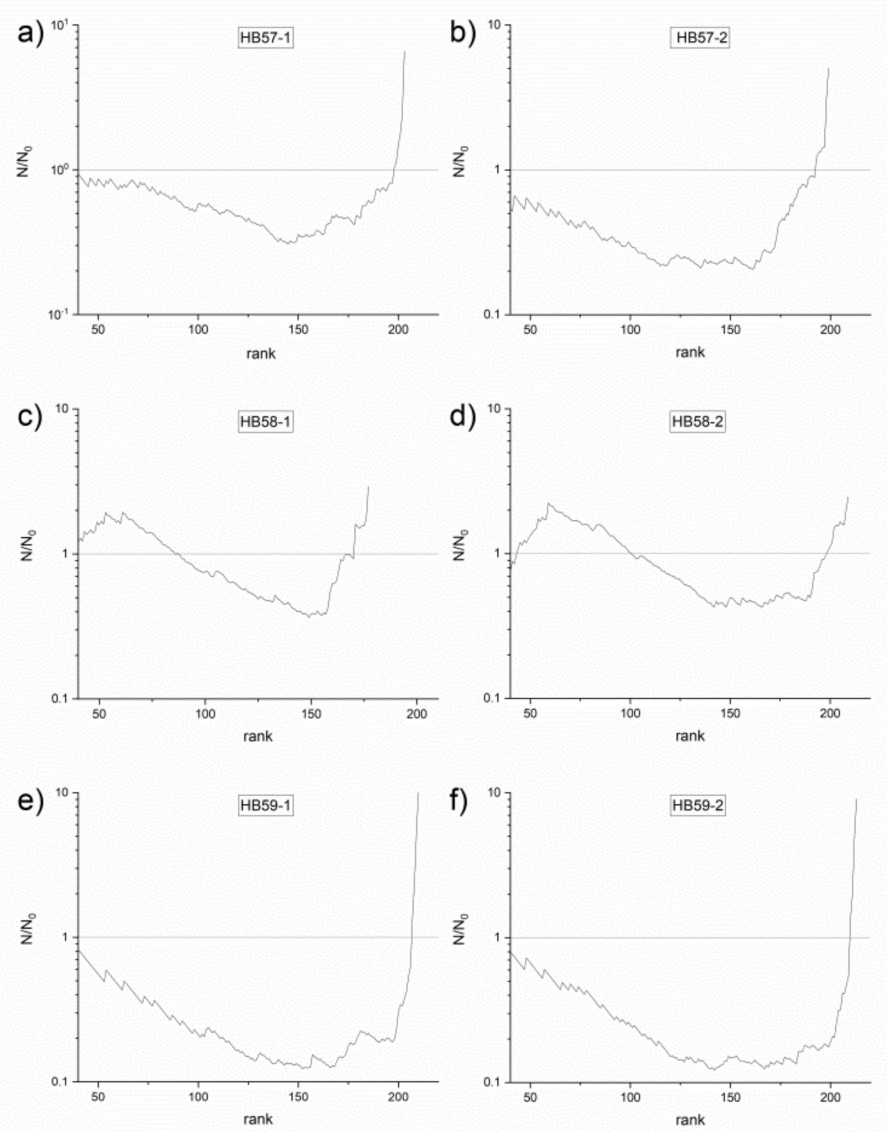

**Figure 4.** Abundance ratio plots in rank order illustrating the deviation of empirically identified abundances from an ideal exponential function with the rank-dependent value $N_0$ related to Equation (2): (**a**) sample HB57-1 (coal seam), (**b**) sample HB57-2 (coal seam), (**c**) sample HB58-1 (bright sediment), (**d**) sample HB58-2 (bright sediment), (**e**) sample HB59-1 (top soil), (**f**) sample HB59-2 (top soil).

The typical OTUs that were present in all samples included *Kryptoniaceae*, *Thermoplasmata*, *Cupriavidus*, and *Candidatus Micrarchaeum* (Figure S3). The *Kryptoniaceae* had particularly high read numbers in the coal seam samples. The phylum *Kryptonia* is associated with high-temperature habitats, in particular geothermal springs [39]. The coal seam samples showed the highest content of *Kryptoniaceae*, whereas this OTU was less abundant in the topsoil. *Thermoplasmata* are also extremophile organisms. They are a group of archaea lacking a cell wall that exhibit both thermophilic and acidophilic properties. Interestingly, one representative was isolated from a coal mine [40]. *Cupriavidus* is known to be a very heavy metal-tolerant genus and is typically found in anthropogenically influenced environments and under harsh conditions [41]. *Microarchaea* have also been recorded in material from a mine; they are found in acidic environments and exhibit both thermophilic and acidophilic properties [42].

In addition to the OTUs mentioned above, the archaea family *Nitrosotaleacea* was highly abundant in the samples from Bennstedt. In contrast to *Kryptonia*, the highest content of *Nitrosotaleacea* was observed in the topsoil samples (Figure S4). *Acidisphaera*,

*Anaerococcus*, and *Micrarchaea* CG1-02-32-21 were highly abundant in the bright sediment but less abundant or absent in the coal seam and the topsoil (Figure 5). In addition to these bacteria, an uncultured genus of *Sulfobacillaceae* and *Haemophilus* were typical of the bright sediment (Figure 6a). By contrast, the OTUs *Candidatus Paracaedibacter*, *Methylovirgula*, *Ehrlichia*, *Azospirillum*, and *Arcobacter* were more abundant in the coal seam but less abundant in the other samples (Figure 6b). *Methylovirgula* is a methylotrophic aerobic bacterium of the family *Beijerinkiaceae,* isolated from beech wood [43]. *Azospirillum* is an aerobic or microaerophilic bacterium typically associated with plant roots and can promote plant growth by nitrogen fixation [44]. Members of the groups *Paracaedibacter* [45] and *Ehrlichia* are known as intracellular parasites [46].

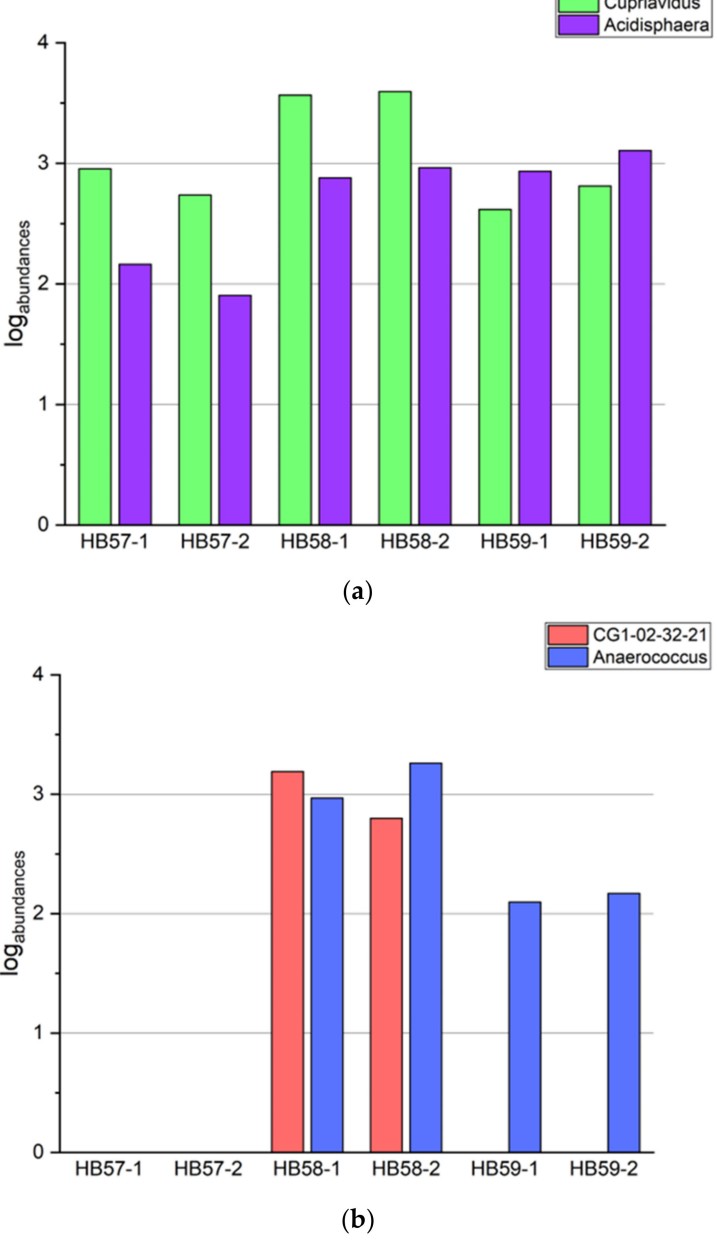

**Figure 5.** Logarithmic abundances (r-values, see Equation (2) in Section 2.4) of special OTUs distinguishing the bacterial communities of the three different sampling sites: (**a**) *Cupriavidus* and *Acidisphaera*; (**b**) *Micrarchaeales CG1-02-32-21* and *Anaerococcus*.

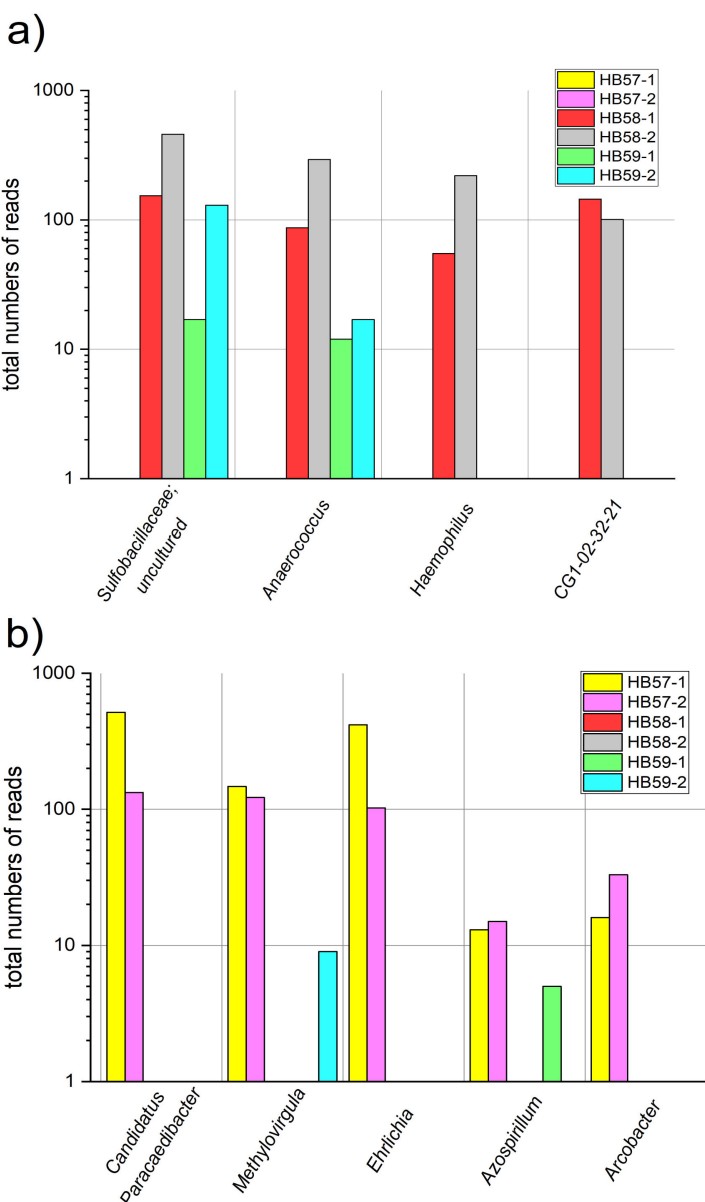

**Figure 6.** Examples of OTUs (absolute read numbers) distinguishing between the soil types (sampling points): (**a**) OTUs exclusively or preferably found in shaft-filling bright sediment; (**b**) OTUs exclusively or preferably found in the coal seam.

The general similarity or dissimilarity between the bacterial communities was well reflected by the correlation plots of OTU abundances. As expected, the sample pairs from the same sampling site showed high similarity in the abundances of each OTU (Figure 7a,b), corresponding to correlation coefficients of 0.913 (HB57-1 and HB57-2), 0.890 (HB58-1 and HB58-2), and 0.9997 (HB59-1 and HB59-2). The correlation coefficients between the different sampling sites (Figure 7c,d) were much lower, e.g., 0.332 between HB57-1 and HB59-1 and 0.324 between HB57-1 and HB59-2 (see also Supplementary Table S1).

The bacterial communities of the coal seam are clearly dominated by the uncultured *Gammaproteobacteria* group KF-JG30-C25 (Figure 8a). This OTU represented 31,914 (HB57-1) and 25,465 reads (HB57-2)—27% and 22% of all reads, respectively (Figure 8a). A further 34,089 reads (29%, HB57-1) and 30,712 reads (27%, HB57-2) concerned the OTUs *Kryptoniaceae*, *Acidobacteria* Subgroup 2, phylum group WP-2, an uncultured genus of *Diplorickettsiaceae*, and *Nitrosomonadaceae* MND1. This means that these six OTUs were responsible for more than half of all DNA reads for the investigated bacterial 16S r-RNA. All

these OTUs were uncultured and thus have not been studied in detail to date. *Kryptoniaceae* are also present in the other samples from Bennstedt but are not very abundant in samples from other places.

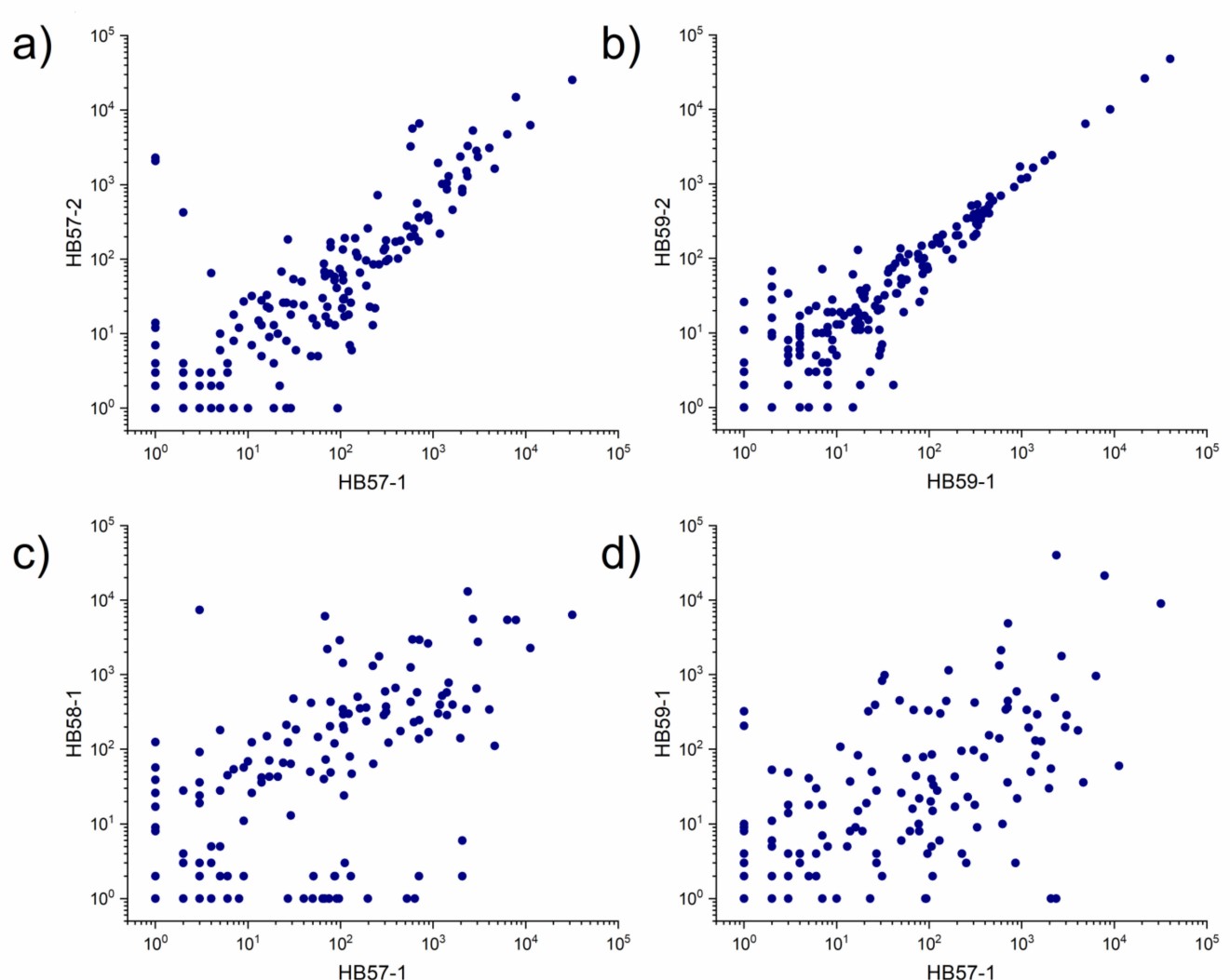

**Figure 7.** Correlation diagrams of OTU abundances for selected pairs of samples. (**a**) correlation of the both coal seam samples HB57-1 and HB57-2, (**b**) correlation of the both samples of bright sediment HB58-1 and HB58-2, (**c**) correlation between the coal seam sample HB57-1 and the bright sediment sample HB58-1, (**d**) correlation between the coal seam sample HB57-1 and the top soil sample HB59-1.

The samples from the bright sediment are dominated by *Nitrosotaleaceae*, with 13,055 (HB58-1) and 16,719 reads (HB58-2) corresponding to 14% and 10% of all reads, respectively (Figure 8b). The *Gammaproteobacteria* group KF-JG30-C25 represented 6.8% and 5.9% of all reads, respectively. The topsoil was dominated by *Nitrosotaleaceae* and *Acidobacteria* subgroup 2. Sample HB59-1 yielded 40,207 reads of *Nitrosotaleaceae* (41% of total reads), and HB59-2 yielded 47,902 reads (41% of total reads). This OTU is probably related to a group of acidophilic ammonia-oxidizing bacteria [47]. The *Gammaproteobacteria* group KF-JG30-C25 represented 9.3% and 8.6% of all reads in the topsoil samples, respectively (Figure 8c).

Besides the abovementioned OTUs, there were several others that were moderately or less abundant but exclusive to samples from one sampling site. These types are shown for all three soil types in Figure 9. There were several OTUs from groups known to be associated with animals in the coal seam samples, including *Ehrlichia*, *Arcobacter*, *Ruminococcus*, *Protoclamydia*, and *Pasteuria* (Figure 9a). *Nitrolancea* is known for its nitrite-oxidizing

activity [48], and *Paludibaculum* [49] and *Pelosinus* [50] are known for their ability to reduce Fe(III). *Deinococcus* is famous for its extraordinarily high radiation resistance [51].

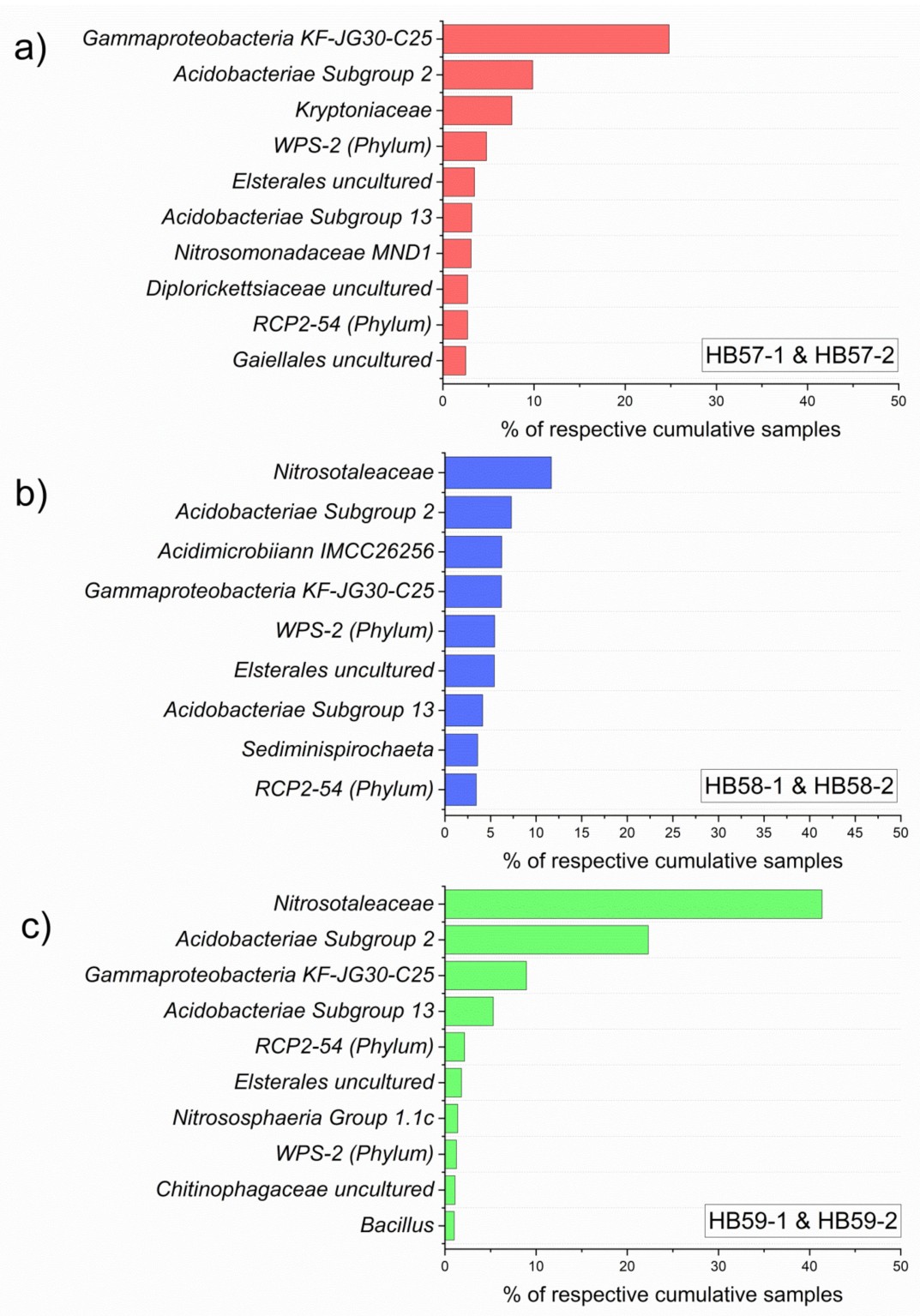

**Figure 8.** Selected species with high abundance. This column chart shows the members of the most dominant OTUs from the three sampling sites: (**a**) coal seam, samples HB57-1 and HB57-2; (**b**) bright sediment, samples HB58-1 and HB58-2; (**c**) topsoil, HB59-1 and HB59-2.

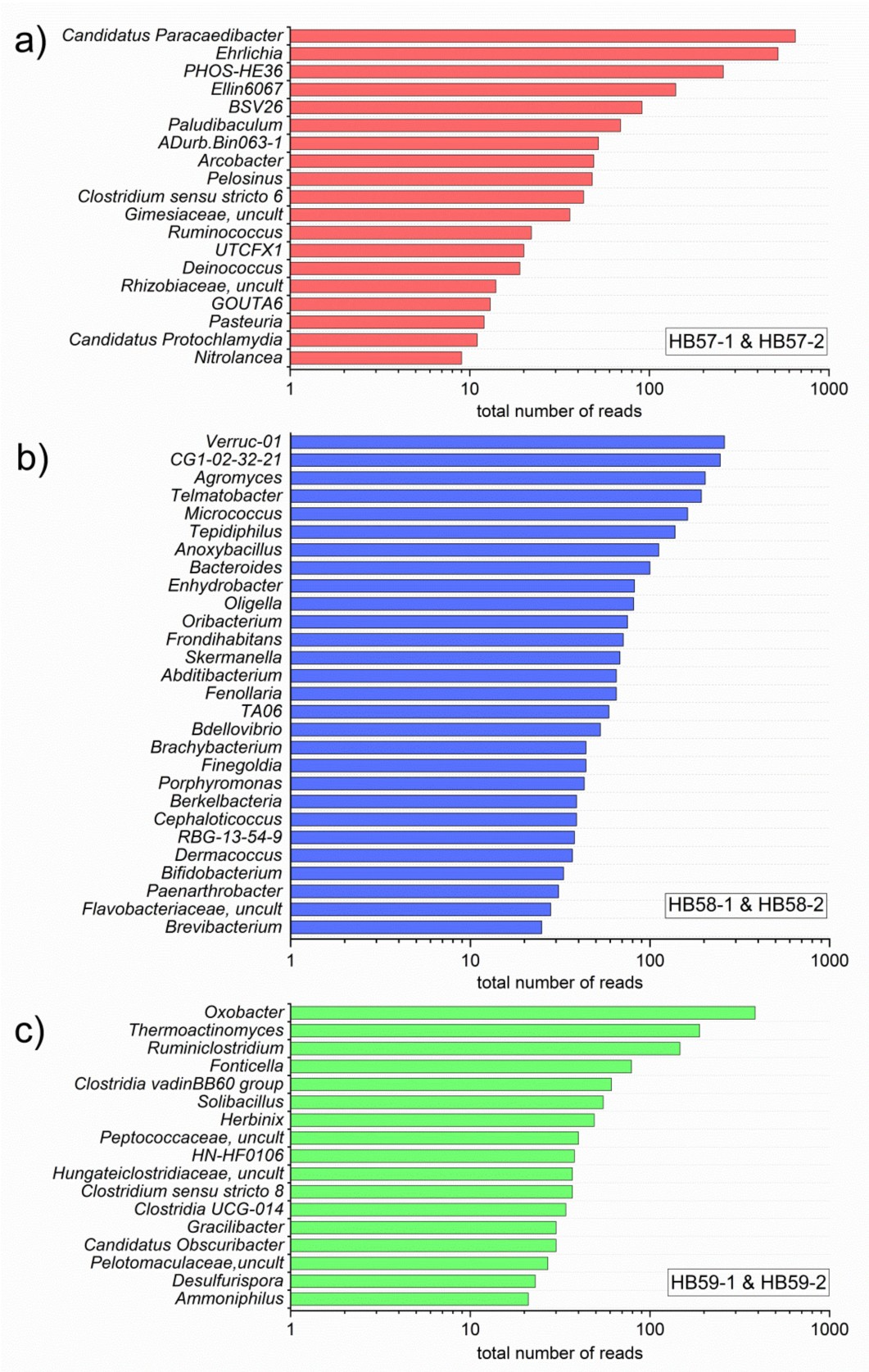

**Figure 9.** Exclusively observed OTUs (number of reads) for the three sampling sites: (**a**) coal seam (HB57-1 and HB57-2), (**b**) bright shaft filling sediment (HB58-1, HB58-2), and (**c**) topsoil sediment (HB59-1, HB59-2). The bars represent the sum of both samples of each sample pair.

The set of OTUs exclusively present in the bright sediment samples was marked by a mixture of anaerobic and aerobic bacteria. Surprisingly, among these OTUs were several bacteria related to species for which a more or less alkaliphilic character has been reported. This fact seems to contradict the low measured pH (about 4.1). Preferred neutral to alkaline growth conditions have been described for *Paenarthrobacter* [52], *Cephaloticoccus* [53], *Frondihabitans* [54], and the thermophilic bacteria *Enhydrobacter* [55] and *Tepidiphilus* [56]. The exclusive appearance of these OTUs seemed to correlate with the special character of the rank order function (Figures 3 and 4) and could have been caused by the former relocation of soil material—possibly associated with a pH change in the environment of these organisms.

Similar to the coal seam samples, a relationship to animals was also described for some exclusively identified OTUs in the bright sediment samples. This concerns *Cephaloticoccus* (an ant gut isolate), *Porphyromonas*, *Brachybacterium* isolated from poultry litter [57]; *Oligella* isolated from urine [58]; and *Haemophilus*, which grows preferentially in blood.

The group of OTUs found exclusively in topsoil (HB59-1 and HB59-2) was characterized by a comparatively high ratio of anaerobic to aerobic bacteria. This could be related to the high content of organic residues in the humus sediment. In particular, strains of the three most abundant OTUs in this group, *Oxobacter* [59], *Ruminiclostridium*, and *Thermoactinomyces* [60,61] are reported to be anaerobic.

Even more than in the other samples, species with reported thermophilic character were found in the topsoil. In addition to *Thermoactinomyces*, these included *Fonticella*, *Herbinix* [62], *Hungateiclostridiaceae* [63], and *Desulfurispora* [64]. *Fonticella* [65] and *Solibacillus* are known to be halotolerant organisms [66,67]. In this regard, *Desulfurispora* [64] is also interesting because of its sulfate-reducing activity, and *Ammoniphilus* is interesting because of its ammonia-dependent metabolism [68].

The different characteristics of the three sampling sites were also well-reflected by the principle component analysis (PCA), considering all OTUs represented with at least 100 (Figure 10a,c) or 10 reads (Figure 10b,d). The sample pairs from the three sampling sites were close to each other for the first two principal components (PC1 and PC2). The high similarity between the two samples from sampling site HB59 (topsoil) was also confirmed by the third and fourth principal components. By contrast, samples HB57-1 and HB57-2 from the coal seam showed high differences in PC3 and PC4, indicating some inhomogeneity in the bacterial community composition. It has to be remarked that samples HB58-1 and HB58-2 were very low in PC2, and samples HB59-1 and HB59-2 were low in PC4. By contrast, the differences in PC3 and PC4 for HB57-1, HB57-2, HB58-1, and HB58-2 were in the same order of magnitude as the differences between the sample groups in PC1 and PC2.

Comparing the three sampling sites, the bright sediment was characterized by the highest number of exclusive OTUs. These samples were distinguished from the others by a group of bacteria related to higher pH. The highest proportion of animal-related species was found in the exclusive OTUs of the coal seam. The exclusive OTUs found in the topsoil were characterized by high proportions of anaerobic and thermophilic species. Conversely, the most abundant OTUs of each sampling site were also highly abundant in the other samples; the groups of exclusively identified OTUs can be regarded as certain "signature-like" bacterial community patterns characteristic of the soil types concerned.

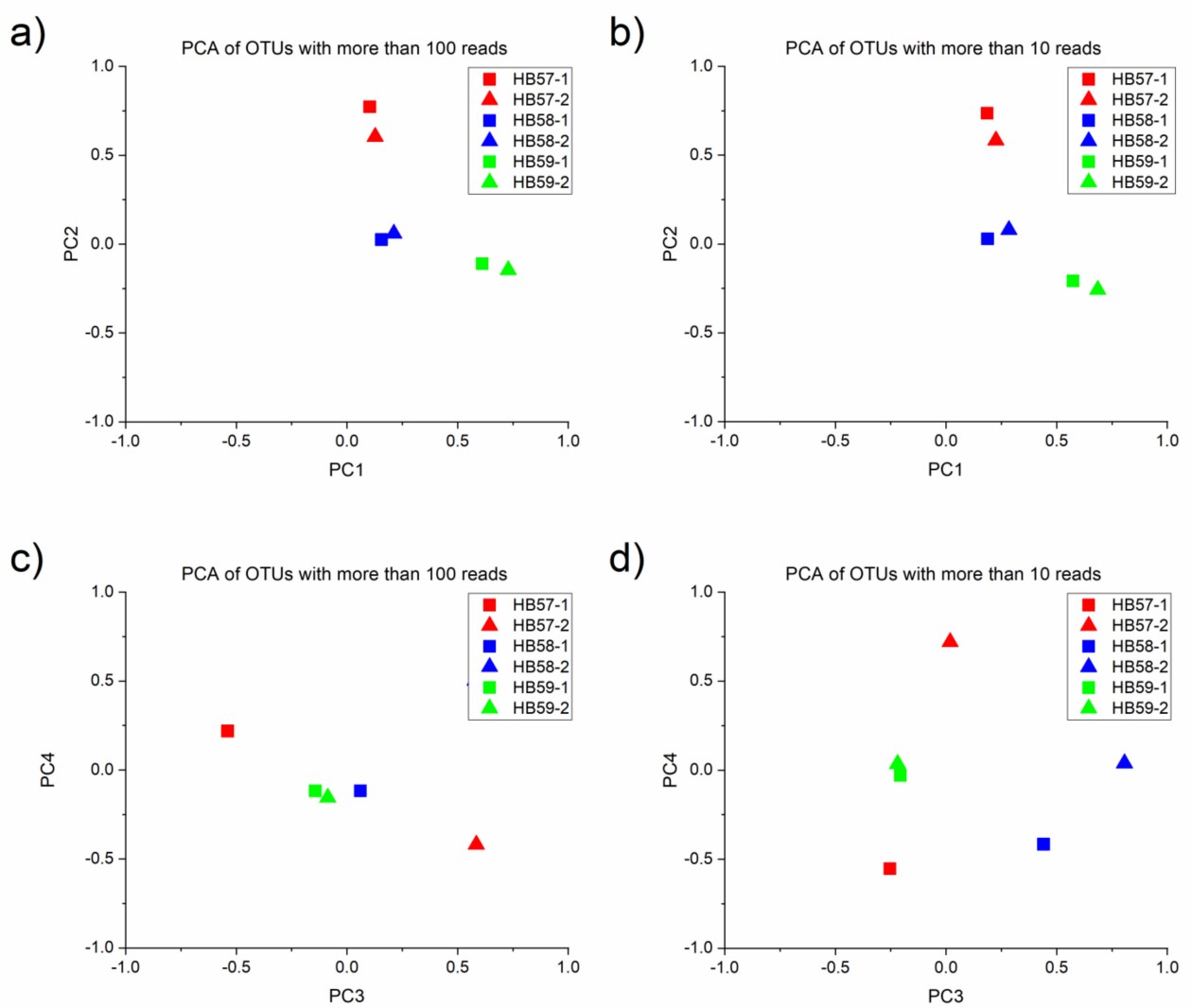

**Figure 10.** PCA for the six investigated samples, HB57-1, HB57-2, HB58-1, HB58-2, HP59-1, andHB59-2, from the three sampling sites (topsoil, coal seam, and shaft-filling sediment). The PCA data are shown for the sets of OTUs with more than 10 and more than 100 reads: (**a**) plot of first and second principle component for OTUs with more than 100 reads, (**b**) plot of first and second principle component for OTUs with more than 10 reads, (**c**) plot of third and fourth principle component for OTUs with more than 100 reads, (**d**) plot of third and fourth principle component for OTUs with more than 10 reads.

## 4. Conclusions

The 16S r-RNA analyses of three investigated sampling sites on the archaeological excavation section in a pre-industrial coal mining area proved the specific characteristics of the contained soil bacterial communities. Although the sampling sites were only a few decimeters apart, the pH of all three sampling sites was very similar, and the electrical conductivity was moderate for all samples, the composition of soil microorganisms showed considerable differences. The specific characteristics of soil bacterial communities corresponded to the different appearances of the soil and the local situations observed during the archaeological excavation.

Two findings were particularly interesting: first, the distribution of OTU abundances, and second, the presence of bacterial species described as alkaliphilic in the bright sediment filling the lower part of the investigated exploration shaft. It is proposed that these observations should be interpreted as a consequence of the fast relocation of soil material during the refilling of the shaft about two centuries ago.

**Supplementary Materials:** The following supporting information can be downloaded at: https://www.mdpi.com/article/10.3390/environments9090115/s1, Figure S1. Excavation site near Bennstedt (Germany). Figure S2. Logarithmic plots of real and idealized rank functions. Figure S3. Abundances of five selected OTUs. Figure S4. Logarithmic abundances (r-values) of two selected OTUs. Table S1. Correlation coefficients for the OTUs of the six samples from Bennstedt.

**Author Contributions:** Conceptualization, L.E. and J.M.K.; methodology, J.M.K.; formal analysis, P.M.G.; investigation, L.E., M.B. and J.M.K.; data curation, P.M.G.; writing—original draft preparation, J.M.K. and L.E.; writing—review and editing, J.M.K. and L.E.; supervision, J.M.K. and J.C.; project administration, J.M.K.; funding acquisition, J.M.K. and L.E. All authors have read and agreed to the published version of the manuscript.

**Funding:** This research was partially funded by Thüringer Landesgraduiertenförderung (Freistaat Thüringen) for L. Ehrhardt.

**Data Availability Statement:** Not applicable.

**Acknowledgments:** We thank Nancy Beetz for assistance with DNA extraction and amplification and Frances Möller for pH and conductivity measurements. L. Ehrhardt is grateful for a scholarship from Thüringer Landesgraduiertenförderung.

**Conflicts of Interest:** The authors declare no conflict of interest.

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
