# Peer review of "Three Soil Bacterial Communities from an Archaeological Excavation Site of an Ancient Coal Mine near Bennstedt (Germany) Characterized by 16S r-RNA Sequencing"

_environments, doi:10.3390/environments9090115_

Round 1

Reviewer 1 Report

Comments and suggestions for the authors The authors submitted a manuscript under the title “Three soil bacterial communities from an archaeological excavation site of an ancient coal mine near Bennstedt (Germany) characterized by 16S r-RNA sequencing”. Overall, the manuscript is demanding. However, it still needs changes and modifications. Here I would like to provide some comments. The manuscript needs to be more concise, and there are many grammar/spelling errors throughout the text. Some sentences are not easy to follow, for example, lines 163-165 (Please write complete word whether it is respectively or something else), line 119 (please check again whether it is 2.5 or 2,5). Please get a native English speaker to correct the whole manuscript as there are a lot of grammatical mistakes, line 88 (Please remove the double space and change the word table 1 into Table 1). Line 236 (change the word table S1 into Table S1). Line 710 (change the word OUT into OTU). Line 903 (change the word OTUS into OTUs). Please improve the structure and design of Table 1. Please explain all figures and graphs properly in the footnote. Graphs should be explained statistically and add a paragraph of statistical analysis in the experimental part. All scientific names need to be italicized in Fig. 5,6,8, S3 and S4. Authors are suggested to design the graphs through good quality software. Adjust the font size and theme properly throughout all figures. Delete page 20 from the manuscript document.

Author Response

Response to Reviewer 1

Comments and suggestions for the authors The authors submitted a manuscript under the title “Three soil bacterial communities from an archaeological excavation site of an ancient coal mine near Bennstedt (Germany) characterized by 16S r-RNA sequencing”. Overall, the manuscript is demanding. However, it still needs changes and modifications. Here I would like to provide some comments.

The manuscript needs to be more concise, and there are many grammar/spelling errors throughout the text. Some sentences are not easy to follow, for example, lines 163-165 (Please write complete word whether it is respectively or something else),

We revised the whole manuscript with focus also on grammar and spelling. Hope it becomes more clearly now.

line 119 (please check again whether it is 2.5 or 2,5). Please get a native English speaker to correct the whole manuscript as there are a lot of grammatical mistakes,

The sentence is corrected, now.

line 88 (Please remove the double space and change the word table 1 into Table 1).

The table is corrected now.

Line 236 (change the word table S1 into Table S1). Line 710 (change the word OUT into OTU).

The table and the related figure caption are corrected, now.

Line 903 (change the word OTUS into OTUs). Please improve the structure and design of Table 1. Please explain all figures and graphs properly in the footnote. Graphs should be explained statistically and add a paragraph of statistical analysis in the experimental part.

The figure caption is corrected, now. The statistical methods are described in more detail in a separate section:

2.4. Data analyses

Different methods are used to evaluate the taxonomic data and sample-specific abundances of OTUs.

First, the percentages of the most abundant phyla are simply compared in a bar chart. The associated plot includes an automatic normalization of the read counts. Bar charts are also used to plot absolute read counts reflecting the different abundances of specific OTUs in various samples.

On the one hand, the correlation of abundances when considering the complete data sets was represented by binary logarithmic correlation plots. They allow comparison of sample pairs based on the individual abundances of all detected OTUs by plotting these abundances with one sample as the x-axis and the other as the y-axis. On the other hand, correlation coefficients were calculated for all sample pairs. The complete table can be found in Table S1 in the Appendix.

Since double logarithmic correlation plots give a better impression of the relationship between pairs of samples than linear plots, normalized logarithmic abundance values were included in some analyses. These normalized abundance r-values are expressed as: the ratio of the individual read counts N to the total read count of a sample Nsum:

r = log10 (1+106 * N / Nsum)                                                                                                       (1)

In these graphs, the complete populations in the bacterial communities are compared according to the magnitude of abundances. In this way, the correlations between higher and lower abundant OTUs are presented in a common picture.

The distribution of OTUs in the different samples can also be investigated by principle component analyses (PCA). The corresponding two- dimensional plots illustrate the similarities and dissimilarities of samples very clearly.

Finally, rank diagrams are also used for a general comparison of the distribution of abundances in each sample. In this approach, all OTUs in a sample are ranked according to their number of reads. The differences in the distribution of OTU abundances can be readily illustrated by these plots, which represent the data sets as normalized rank plots.

The difference between the light sediment communities and the other samples is illustrated by rank difference plots between the real and a theoretical ideal distribution of abundances. Therefore, the logarithmic rank plots are approximated by a linear function reflecting the over-all exponential character of the abundance distribution. This approximation is given by the following equation:

N0 = e a * k                                           or                           ln (N0) = a * k                    (2)

where N0 describes the (theoretical) number of reads for each OTU corresponding to its position k in the rank order. The coefficient a depends on the ratio of the total number of reads and OTUs.

A PCA analyses was integrated, additionally, in order to compare samples. Therefore, an additional paragraph is introduced:

The different character of the three sampling sites is also well reflected by the Principle Component Analysis (PCA) reconsidering all OTUs which are represented by 100 reads or more (Fig. 9a, c) or by at least 10 reads (Fig. 9b, d). The pairs of samples from the three sampling sites are located close together for the first two principle components (PC1 and PC2). The high similarity between the both samples from sampling site topsoil is also confirmed by the third and fourth principle component. In contrast, the both samples HB57-1 and HB57-2 from sampling site coal seam show high differences in PC3 and PC4 speaking for a certain inhomogeneity in the composition of the bacterial community.

and a new Fig. 10 (PCA graphs) is added.

All scientific names need to be italicized in Fig. 5, 6, 8, S3 and S4.

It is corrected now.

Authors are suggested to design the graphs through good quality software. Adjust the font size and theme properly throughout all figures.

The diagrams were created again, with a better resolution.

Delete page 20 from the manuscript document.

Reviewer 2 Report

The manuscript reports 16S rRNA sequencing analysis of soil samples from an archaeological area of ancient brown coal mining area. The phyla were typical for soil samples, but a decreasing number of reads in the rank order was interpreted as less active or dormant bacteria and they might have been more active in the past. The data was analyzed throughly for typical OTUs which are present in all samples, Kryptoniaceae showed particular high read numbers in the coal seam samples. The coal seam samples showed the highest 213 content of Kryptoniaceaea, but less abundant in the top soil. Similarly, the archaeal family Nitrosotaleacea is highly abundant in the samples. The methods were presented in detail and results were disscussed to support the main conclussion “as a consequence of the fast relocation of soil material during the re-filling of the shaft about two centuries ago”. I think this is well-written article with an interesting result.

Author Response

Thank you very much for the positive judgement of our work!

Reviewer 3 Report

The manuscript entitled "Three Soil Bacterial Communities from an Archaeological Excavation Site of an Ancient Coal Mine near Bennstedt (Germany) Characterized by 16S r-RNA Sequencing" should have the potential to be of great interest to the scientific field, but I have serious comments about it.

Sampling alone was unrepresentative. It is necessary to have at least 3-4 repetitions within 1 site and not only 2. This is insufficient. I appreciate the choice of 3 types of sites (topsoil, bright sediment, and coal seam). Still, the number of repetitions should have been higher and taken at a more considerable horizontal distance so that the scope of the given site was as high as possible. If the 2 replicates were taken very closely together, then the results of the analysis of these 2 samples cannot be considered to be characteristic overall for the given locality. The chemical specification of the soil, sediment, i.e., from which the mgDNA was isolated, is missing. It is not clear what type of soil it was and what the elemental composition of at least the essential chemical elements was. It is not enough to state only the pH value and electrical conductivity.

What concentration of DNA was used in PCR? The amount of DNA is not clear from this volume. The methodology needs to be described in more detail and the chapter on statistical evaluation is completely missing. In what program were the correlation coefficients made? What data did they use?

I do not understand why the sampling points were evaluated separately. The data from those 2 repetitions within 1 sampling site should be pooled and the standard deviations or standard errors within these values should be plotted.

Indication of the formulas according to which the conversion to Figures S2, S3 or Fig. 4 should be mentioned in the methodological part and not in the results. Where do those formulas come from? There is also no description of the marks in the given equations.

Lines 231-236 - The pairs shown do not correspond to Figure 7. Figure 7 shows completely different combinations of pairs, with Figures 7a and 7c having the same x and y axis name.

Lines 238-251 - I don't know which figure this text links to. I can't verify that.

It is necessary to better describe Figure caption. It must be stated which samples come from which sampling point.

Figure 3 - I can't assign colour curves to samples.

I would also recommend evaluating the results with some multivariate statistical methods. Such an evaluation of the results is confusing.

Author Response

Reviewer 3

The manuscript entitled "Three Soil Bacterial Communities from an Archaeological Excavation Site of an Ancient Coal Mine near Bennstedt (Germany) Characterized by 16S r-RNA Sequencing" should have the potential to be of great interest to the scientific field, but I have serious comments about it.

Sampling alone was unrepresentative. It is necessary to have at least 3-4 repetitions within 1 site and not only 2. This is insufficient. I appreciate the choice of 3 types of sites (topsoil, bright sediment, and coal seam). Still, the number of repetitions should have been higher and taken at a more considerable horizontal distance so that the scope of the given site was as high as possible. If the 2 replicates were taken very closely together, then the results of the analysis of these 2 samples cannot be considered to be characteristic overall for the given locality.

Thank you for this advice! We agree completely with the demand for a higher number of samples from the same sampling site. Unfortunately, for this study no more samples are available and a new sampling is not possible. It has to be taken in mind that the number of possible samplings might be dependent on the specific archeological situation.  In future investigations with new sampling programs, we will include as many samples as possible. For the recent study we try to exploit the obtained data as well as possible.

The chemical specification of the soil, sediment, i.e., from which the mgDNA was isolated, is missing. It is not clear what type of soil it was and what the elemental composition of at least the essential chemical elements was. It is not enough to state only the pH value and electrical conductivity.

It is true that a detailed chemical specification of soil samples is of large interest, but it was not possible to get these data in the frame of the recent study. With respect to the fact that salt content and deviations from the neutral situation are particular important for soil bacterial communities, we decided to measure electrical conductance and pH in order to can classify the samples by these very important parameters and allowing a comparison with soil bacterial communities from other investigations.

What concentration of DNA was used in PCR? The amount of DNA is not clear from this volume. The methodology needs to be described in more detail and the chapter on statistical evaluation is completely missing. In what program were the correlation coefficients made? What data did they use?

The initial amount of DNA could be estimated only approximately by gel electrophoreses of the amplification product.

The statistical evaluation was improved;  please see below: new section: 2.4. Data analyses

I do not understand why the sampling points were evaluated separately. The data from those 2 repetitions within 1 sampling site should be pooled and the standard deviations or standard errors within these values should be plotted.

The separate evaluation gives a mode detailed impression about the homogeneity or inhomogeneity of local compositions of bacterial communities. For example, such impressions are get by the graphs of PCA which are included in the revised version of the manuscript, now (see also below).

Indication of the formulas according to which the conversion to Figures S2, S3 or Fig. 4 should be mentioned in the methodological part and not in the results. Where do those formulas come from? There is also no description of the marks in the given equations.

The statistical methods are described in more detail in a separate section:

2.4. Data analyses

Different methods are used to evaluate the taxonomic data and sample-specific abundances of OTUs.

First, the percentages of the most abundant phyla are simply compared in a bar chart. The associated plot includes an automatic normalization of the read counts. Bar charts are also used to plot absolute read counts reflecting the different abundances of specific OTUs in various samples.

On the one hand, the correlation of abundances when considering the complete data sets was represented by binary logarithmic correlation plots. They allow comparison of sample pairs based on the individual abundances of all detected OTUs by plotting these abundances with one sample as the x-axis and the other as the y-axis. On the other hand, correlation coefficients were calculated for all sample pairs. The complete table can be found in Table S1 in the Appendix.

Since double logarithmic correlation plots give a better impression of the relationship between pairs of samples than linear plots, normalized logarithmic abundance values were included in some analyses. These normalized abundance r-values are expressed as: the ratio of the individual read counts N to the total read count of a sample Nsum:

r = log10 (1+106 * N / Nsum)                                                                                                       (1)

In these graphs, the complete populations in the bacterial communities are compared according to the magnitude of abundances. In this way, the correlations between higher and lower abundant OTUs are presented in a common picture.

The distribution of OTUs in the different samples can also be investigated by principle component analyses (PCA). The corresponding two- dimensional plots illustrate the similarities and dissimilarities of samples very clearly.

Finally, rank diagrams are also used for a general comparison of the distribution of abundances in each sample. In this approach, all OTUs in a sample are ranked according to their number of reads. The differences in the distribution of OTU abundances can be readily illustrated by these plots, which represent the data sets as normalized rank plots.

The difference between the light sediment communities and the other samples is illustrated by rank difference plots between the real and a theoretical ideal distribution of abundances. Therefore, the logarithmic rank plots are approximated by a linear function reflecting the over-all exponential character of the abundance distribution. This approximation is given by the following equation:

N0 = e a * k                                           or                           ln (N0) = a * k                    (2)

where N0 describes the (theoretical) number of reads for each OTU corresponding to its position k in the rank order. The coefficient a depends on the ratio of the total number of reads and OTUs.

A PCA analyses was integrated, additionally, in order to compare samples.

Lines 231-236 - The pairs shown do not correspond to Figure 7. Figure 7 shows completely different combinations of pairs, with Figures 7a and 7c having the same x and y axis name.

Thank you for this important advice! Indeed the axis was wrong in Fig. 7a. We apologize for that. The figure is corrected now.

Lines 238-251 - I don't know which figure this text links to. I can't verify that.

The content of the related § was not illustrated in the first submitted manuscript. An additional figure (Fig. 8) is included, now, for displaying the abundances of the dominating OTUs in the single samples.

It is necessary to better describe Figure caption. It must be stated which samples come from which sampling point.

Sample names are marked in the archaeological section image and a description is given of what type of sample it is (sediment, topsoil, or coal seam).

Figure 3 - I can't assign colour curves to samples.

Fig. 3 was modified in order declare the meaning of the color lines.

I would also recommend evaluating the results with some multivariate statistical methods. Such an evaluation of the results is confusing.

Following this valuable advice, we include now Principle Component Analyses (PCA) for evaluating the data on bacterial populations from the three different sampling sites. The results are discussed and the graphs are shown in Fig 10.

Round 2

Reviewer 3 Report

The manuscript has been significantly corrected and I realise that at this stage, nothing can be done about the sampling itself, which I consider the biggest obstacle to acceptance. If the editor has no problem with that, I can accept your statement about this.  On the other hand, I still have a few minor comments:

Lines 185-186 + 249 – The word "significantly" is used when something is statistically confirmed. Were these amounts regarding Myxococcota and Verrucomicrobia statistically significant? If so, it is necessary to indicate the statistical method used for the given statement and the significance level.

Lines 251-253 + 282 – Citations supporting these sentences should be used.

Figure 6 caption is missing.

Figure 10 – Please add the percentage corresponding to PC1 - PC4. If the % for PC3 and PC4 are much lower than for PC1 and PC2, they are not very informative for the distribution of individual locations.

Author Response

Response to reviewer:

Lines 185-186 + 249 – The word "significantly" is used when something is statistically confirmed. Were these amounts regarding Myxococcota and Verrucomicrobia statistically significant? If so, it is necessary to indicate the statistical method used for the given statement and the significance level.

Line 185-186: “significantly” was deleted.

Line 249: The sentence was modified:

“In contrast, the OTUs Candidatus Paracaedibacter, Methylovirgula, Ehrlichia, Azospirillum and Arcobacter are higher abundant in the coal seam, but less in the other samples (figure 6b).”

Lines 251-253 + 282 – Citations supporting these sentences should be used.

The missing references are added now:

44 Tarrand, J.J.; Krieg.N.; Dobereiner, J. Taxonomic study of Spirillum-lipoferum group, with description of a new genus, Azospirillum gen-nov and 2 species, Azospirillum-lipoferum (Beijerinck) comb nov. and Azospirillum-brasilense sp-nov. Can J. Microbiol. 1978, 24, 967-980.

45 Midha, S.; Rigden, D.J.; Siozios, S.; Hurst, G.D.D.; Jackson, A.P. Bodo saltans (Kinetoplastida) is dependent on a novel Paracaedibacter-like endosymbiont that possesses multiple putative toxin-antitoxin systems. ISME J. 2021, 15, 1680-1694.

46 Lu, M.; Tang, G.P., Ren, Z.Q.; Zhang, J.; Wang, W.; Qin, X.C.; Li, K. Ehrlichia, Coxiella and Bartonella infections in rodents from Guizhou Province, Southwest China. Ticks and Tick-borne Diseases 2022, 13, 101974.

51 Mattimore, V.; Battista, J.R. Radioresistance of Deinococcus radiodurans: Functions necessary to survive ionizing radiation are also necessary to survive prolonged desiccation. J. Bacteriol. 1996, 178, 633-637.

Figure 6 caption is missing.

Sorry about that! The missing figure caption is added , now

Figure 10 – Please add the percentage corresponding to PC1 - PC4. If the % for PC3 and PC4 are much lower than for PC1 and PC2, they are not very informative for the distribution of individual locations.

The PCA-diagrams are not further scaled. Thus, the numbers for PC3 and PC4 are mostly in the same order of magnitude as the numbers for PC1 and PC2. But, it has to be remarked, that the samples HB58-1 and HB58-2 are very low in PC2, the samples HB59-1 and HB59-2 low in PC4. For clarifying that the related paragraph was completed by following:

“…In contrast, the two samples HB57-1 and HB57-2 from the coal seam show high differences in PC3 and PC4 indicating some inhomogeneity in the bacterial community composition. It has to be remarked that the samples HB58-1 and HB58-2 are very low in PC2, the samples HB59-1 and HB59-2 low in PC4, whereas the differences in PC3 and PC4 for HB57-1, HB57-2, HB58-1 and HB58-2 are in the same order of magnitude as the differences between the sample groups in PC1 and PC2.”